



# Retrieval of Microphysical Parameters of Monsoonal rain Using X-band Dual-polarization Radar: Their Seasonal Dependence and Evaluation

**Kumar Abhijeet[1,2], T. Narayana Rao*[1], N. Rama Rao[2], and K. Amar Jyothi[3]**

[1]National Atmospheric Research Laboratory, Gadanki-517112, Andhra Pradesh, India

[2]Indian Institute of Space Science and Technology, Thiruvananthapuram-695547, Kerala, India

[3]National Centre for Medium-range Weather Forecast, Noida-201309, India

*Correspondence to*: T. Narayana Rao (tnrao@narl.gov.in)

**Abstract:** Multiyear measurements from Joss-Waldvogel disdrometer (5 years) and X-band dual-polarization radar (2 years) made at Gadanki (13.5 N, 79.18 E), a low latitude station, are used to i) retrieve appropriate raindrop size distribution (DSD) relations for monsoonal rain, ii) understand their dependency on temperature, raindrop size-shape model and season and iii) assess polarimetric radar DSD retrievals by various popular techniques (Exponential-Exp, Constrained Gamma – CG, Normalized Gamma – N-Gamma and $\beta$ methods). The coefficients obtained for different DSD relations for monsoonal rain are found to be

different from that of existing relations elsewhere. The seasonal variation in DSD is quite large and significant and as a result the coefficients also vary considerably between the seasons. The slope of the drop size - shape relation, assumed to be constant in several studies, vary considerably between the seasons with warmer seasons showing smaller slope value than cold season. It is found that the constant (0.062) used in linear drop shape models is valid only for cold season. The derived coefficients for CG method for different seasons coupled with those available in the literature reveals that the warm seasons/regions typically have

larger curvature and slope values than in cold seasons/regions. The coefficients of mass weighted mean diameter ($D_m$) – differential reflectivity ($Z_{DR}$) exhibit strong dependency on drop shape model, while those for the derivation intercept parameter exhibit strong seasonal dependency. Using the retrieved relations and X-band polarimetric radar at Gadanki, four popular DSD methods are evaluated against disdrometer measurements collected over 12 events. All the methods estimated $D_m$ reasonably well with small root mean square error, however failed to estimate intercept parameter accurately.  Only N-gamma method estimated the

normalized intercept parameter reasonably. Problems associated with specific differential phase ($K_{DP}$)-based estimates close to the radar location, particularly during overhead convection, are also discussed.

## 1. Introduction

Raindrop size distribution (DSD) is the fundamental property of precipitation and its space-time variability depends on a variety of microphysical and dynamical processes inside and below the clouds (Radhakrishna and Rao, 2009; Rao et al., 2009; Rosenfeld and Ulbrich, 2003).  Such information is crucial even for numerical weather prediction models as these microphysical processes

are fundamental blocks in microphysical schemes (Gao et al., 2011). Knowledge of DSD is not only required for fundamental understanding of microphysical processes, but also for a variety of operational applications in the fields of hydrology, meteorology, agriculture, and road transportation sectors, among others (Rosenfeld and Ulbrich, 2003; Serio et al., 2019; Uijlenhoet, 2001, and



references therein). Disdrometers provide this crucial information continuously, but only at the Earth's surface. Radars, on the
other hand, provide DSD both in space and time and, therefore, play a major role in improving our understanding on microphysical
processes in a variety of precipitating systems (Ryzhkov and Zrnic, 2019).

Remarkable progress has been made in the polarimetric (dual-polarization) radar technology and their utilization for research and
operational applications in the recent past (Bringi and Chandrasekar, 2001; Rauber and Nesbitt, 2018; Ryzhkov et al., 2022;
Ryzhkov and Zrnic, 2019). Besides improving the rain rate estimation, the polarimetric radars offer unique information on
microphysical properties of precipitation, like the DSD (Anagnostou et al., 2008a; Cao and Zhang, 2009; Gorgucci et al., 2001;
Koffi et al., 2014; Maki et al., 2005; Moisseev and Chandrasekar, 2007; Penide et al., 2013; Seliga and Bringi, 1976; Zhang et al.,
2001). They also provide information on the shape, orientation and phase state of hydrometeors, by employing sophisticated
hydrometeor classification algorithms, like fuzzy logic and Bayesian classification (Liu and Chandrasekar, 2000; Marzano et al.,
2007; Vivekanandan et al., 1999; Zrnic et al., 2001). Several earlier studies demonstrated that the DSD parameters can be used not
only to understand the microphysics of precipitation and clouds, but also for improved rain rate estimation ( Zhang et al., 2001;
Gorgucci et al., 2001; Vivekanandan et al., 2003; Vulpiani et al., 2006; Brandes et al., 2004a; Cao et al., 2010, 2008; Gosset et al.,
2010; Anagnostou et al., 2013; Koffi et al., 2014; Ryzhkov and Zrnic, 2019; ).  They have shown that DSD-based rain rate
estimation outperforms the fixed power-law rainfall estimation from reflectivity fields and it is equivalent to those derived with
multi-parameter retrievals of rainfall with polarimetric radars (Anagnostou et al., 2010; Brandes et al., 2003; Vivekanandan et al.,
50  2003).

Earlier studies followed various approaches to retrieve the DSD from polarimetric radars: statistical techniques and physics-based
empirical relations between DSD model parameters and polarimetric products. Statistical methods, include neural network
(Vulpiani et al., 2006), Bayesian (Cao et al., 2010) and different variants of Bayesian, like variational methods (Cao et al., 2013;
Yoshikawa et al., 2016), find the non-linear relationships between DSD and polarimetric parameters making use of mathematical
techniques. These methods either train the chosen model or build a priori database using existing information, which then will be
used to retrieve DSD parameters. Physics-based methods assume that the DSD follows some functional form (exponential, gamma
or normalized gamma) and derive relation between DSD model parameters and polarimetric radar parameters empirically.
Different methods evolved over the years since (Seliga and Bringi, 1976)'s exponential method (Exp), including constrained
gamma (CG) (Zhang et al., 2001), Beta ($\beta$) (Gorgucci et al., 2000), normalized gamma (N-Gamma) (Bringi et al. 2002; Anagnostou
et al., 2008a; Tokay et al., 2020a), double-moment model (Raupach and Berne, 2017), self-consistent with optical parameterization
attenuation correction and microphysics estimation (*SCOPE-ME*) (Anagnostou et al., 2009) and inverse model (Alcoba et al., 2022;
Wen et al., 2018).

Among the above methods, the Exp, CG, N-Gamma and $\beta$ methods are extensively used by researchers. The two-parameter
exponential model assumes that the distribution of rain drops follows an exponential form and its parameters can be retrieved from
two polarimetric measurements, namely horizontal reflectivity factor ($Z_H$) and differential reflectivity ($Z_{DR}$) (Seliga and Bringi,
1976). The CG method assumes that the DSD follows gamma distribution (Ulbrich, 1983) and the retrieval of three gamma
parameters is achieved using two independent polarimetric measurements and an empirically derived constrained relation between
shape ($\mu$) and slope ($\Lambda$) parameters of gamma distribution (Brandes et al., 2004a; Zhang et al., 2001). The $\beta$ method follows
normalized DSD concept, described in (Willis, 1984; Illingworth and Blackman, 2002; Testud et al., 2001). Here, the DSD is
normalized with respect to liquid water content, which allows studying variations in DSD shape by accounting variations of water
content. In addition, this method considers raindrop shape – diameter relation as a variable (Gorgucci et al., 2001), instead of a





fixed relation for equilibrium shape of a raindrop (Pruppacher and Beard, 1970). The $Z_H$, $Z_{DR}$ and Specific differential phase ($K_{DP}$) are used to obtain the slope ($\beta$) of the above relation, which intrinsically considers changes in drop oblateness that increases with the size of a raindrop.

Earlier studies derived/generated several empirical relations relating polarimetric variables at different frequencies to obtain the DSD parameters. Some of these relations are obtained from simulations or parameterizations and the others from observations (Adirosi et al., 2020; Anagnostou et al., 2008a, 2008b; Brandes et al., 2004; Gorgucci et al., 2001; Maki et al., 2005; Rao et al., 2006; Seliga and Bringi, 1976; Tang et al., 2014; Tokay et al., 2020; Zhang et al., 2001 and references therein). Unfortunately, the above relations are found to be quite different at different locations due to large DSD variations (Brandes et al., 2004b; Chen et al., 2017; Chu and Su, 2008; Kim et al., 2020; Kumar et al., 2011; Rao et al., 2006; Seela et al., 2018; Tang et al., 2014; Zhang et

al., 2001; Zheng et al., 2020). Not only between regions, the DSD and $\mu$-$\Lambda$ relation are also found to vary between different regimes (i.e., eye wall and rain bands) of a cyclone (Bao et al., 2020). These variations are caused primarily by different prevailing atmospheric conditions (in different geographical regions), in which the drop forms and the DSD evolves (Lee and Zawadzki, 2005). The above reported relations are based on the data from America, Japan, Taiwan, Singapore and China and, therefore, are

more appropriate for the above regions, while such relations do not exist for India (barring one study by (Rao et al., 2006) using a limited dataset). The first objective of this paper is to derive suitable DSD retrieval relations at X-band for monsoonal rainfall over Indian region, where several X-band polarimetric radars are either installed or being installed. An X-band dual-polarization radar (DROP-X – Dual polarization Radar for Observing Precipitation at X-band), developed indigenously, recently became operational at Gadanki (13.5 N, 79.18 E) (Rao et al., 2022).

It is also known from earlier studies that the DSD varies not only with the climatic regime, but also with the season at the same location. For example, the DSD at a single station can be influenced by both the oceanic and continental systems, depending on the wind and circulation patterns (Kozu et al., 2006; Radhakrishna and Rao, 2009; Rao et al., 2009, 2001; Tokay et al., 2002). Recently, (Rao et al., 2018) noted large differences in coefficients of attenuation correction relations in different seasons. Given such large variability in DSD from one season to the other in the southeastern peninsular India, one should also examine the impact

of the observed seasonal variation on DSD retrieval methods. This forms the second objective of this manuscript.

There have been differences of opinion over the validity of the retrieval of above relations ($\mu$-$\Lambda$ relation and $\beta$ method), usage of DSD models (exponential vs gamma vs normalized gamma) and on drop shape-size relations (linear and constant vs linear but variable vs. polynomial). Earlier, a few studies compared different DSD retrieval techniques (Anagnostou et al., 2008b, 2008a; Brandes et al., 2006, 2004a; Tokay et al., 2020b; Zhang et al., 2006). Such efforts were not made for monsoonal rain. Given the

large seasonal variability in DSD, it is important to evaluate such schemes using observations from polarimetric radars. The present study, therefore, evaluates the retrieved mass weighted mean diameter ($D_m$) and intercept parameter ($N_0$) or normalized intercept parameter ($N_w$) of DSD from DROP-X measurements and derived relations.

The remainder of this paper is organized as follows. Section 2 describes instruments, data and methodology (scattering simulations, deriving polarimetric products and DSD models) used in the present study. Relations between polarimetric products and

exponential/gamma model parameters are empirically derived in Section 3. Seasonal dependence of coefficients of the above relations and their variation with temperature are also discussed in Section 3. The retrieved DSD parameters from radar measurements are evaluated against independent reference dataset in Section 4. Section 5 summarizes important findings from the present study.





## 2. Data and Methodology

### 2.1. Data and Instrumentation

Measurements from DROP-X and collocated Joss-Waldvogel disdrometer (JWD) at National Atmospheric Research Laboratory (NARL), Gadanki are used in the present study. Gadanki is located in a complex hilly terrain of varying heights in the range of 200-500 m above the ground level. It is located in southeast India and experiences rain in 3 seasons. The southwest monsoon (SWM-June through September) is the main monsoon season in which it receives ~53% of its annual rainfall. This region also receives considerable rainfall (35% of annual rainfall) during the northeast monsoon (NEM – October through December) and the remaining annual rainfall occurs during the premonsoon season (PRE – March through May) (Rao et al. 2009; Radhakrishna and Rao, 2021). The rainfall is predominantly convective in nature (53.3% of total rainfall), while stratiform rain (30.2%) and shallow rain (16.6%) contributes considerably (Rao et al., 2008; Saikranthi et al., 2014).

The DROP-X was developed indigenously by Radar Development Area (RDA) of ISRO Telemetry, Tracking and Command Network (ISTRAC) and NARL. The radar is placed on top of a building of 13 m height constructed on a small hillock to minimize blockages due to the local canopy. The DROP-X operates in the frequency range of 9.33-9.34 GHz and has two independent channels for transmission and reception for horizontal and vertical polarized signals. It is equipped with two solid-state transmitters with a peak power of 300 W, one each for each polarization. Other important specifications of the radar are given in Table 1. For the present study, measurements made during 2019 and 2020 are utilized. During the above period, the DROP-X was operated in regular plan position indicator (PPI) mode with a revolution speed of 2 revolutions per minute (rpm) and in 10 elevations (1°-10° with an interval of 1°). Each volume scan takes ~6 min.

**Table 1: Important specifications of DROP-X**

| S. No. | Parameters | Specifications |
|--------|-----------|----------------|
| 1. | Weather radar | Polarimetric type |
| 2. | Transmitter type | Solid state power amplifier module |
| 3. | Operating frequency | 9.33 – 9.34 GHz |
| 4. | PRF | 825 & 1500 Hz |
| 5. | Max. range capability | 150 km |
| 6. | Pulse width | 0.5μs, 16μs and 128μs |
| 7. | Peak output power | 300(H)/300(V) |
| 8. | Wave form | NLFM |

The JWD (RD-80) at Gadanki, used in the present study, is an impact type disdrometer that records the number of rain drops hitting the 50 cm$^2$ surface of the sensor. It can identify 128 sizes of rain drops with diameters ranging from 0.3 to 5.4 mm and later arranges the data collected in 1 minute in 20 drop size channels. All rain integral parameters like reflectivity ($Z$), rainfall rate ($R$), and $D_m$ are estimated directly from the measured DSD, using standard formulae (Rao et al., 2001). The measurements were corrected for dead time of the instrument (Sheppard and Joe, 1994). Five years (2016-2020) of JWD measurements were used in the present study. First three years of data are used to obtain coefficients of the relations between polarimetric radar measurements and geophysical parameters. Few quality checks have been performed to retain good quality data. The data are considered to be





valid only when $R$ is greater than 0.5 mm hr$^{-1}$ and available in at least 4 continuous drop size channels of disdrometer. A total of 26,449 minutes of DSD data satisfied the above quality checks and are used in the present study. The latter two years of data are also subjected to the above quality checks and then are used to evaluate the performance of DSD retrievals with DROP-X. The disdrometer is located ~200 m away from the radar location and at an azimuth angle of 77.5°. To match radar temporal resolution,

disdrometer data are averaged over 6 minutes. The radar measurements around the disdrometer are also averaged to obtain statistically robust estimate. For averaging, data of 3 range bins each in 3 azimuthal directions centered around disdrometer location and in 3 elevation angles (4°, 5° and 6°) are utilized (i.e., a volume averaging of 450 m x 10.5 m x 10.5 m at a height 17 m above the disdrometer). The elevation angles are chosen in such a way that the targeted volume is as close as possible to the reference disdrometer, but not contaminated by the ground clutter.

**2.2. Methodology to retrieve polarimetric parameters**

The scattering and extinction amplitudes are calculated using *T*-matrix scattering simulations (Mishchenko et al., 1996). Following raindrop size-shape models and parameters are used for these simulations. Scattering amplitudes are computed at 9.34 GHz frequency with four standard raindrop size - shape models, i.e., (Pruppacher and Beard, 1970; Beard and Chuang, 1987; Andsager et al., 1999; Brandes et al., 2002). Though simulations with (Andsager et al., 1999) model are finally used for further analysis,

simulations with other models are performed to check the dependency of scattering amplitudes and retrieved polarimetric radar parameters on drop shape model. The axis ratio is assumed to be the same as that given by the above drop shape models. Since (Brandes et al., 2002) model has accounted the effect of raindrop oscillations in their axis ratio, no additional canting angle distribution is considered when it is used in simulations. For simulations with other drop shape models, Gaussian canting angle distribution with a mean of 0° and a standard deviation of 10° is considered. Simulations are performed at different environmental

temperatures, from 0°C to 30°C with an interval of 5°C, to understand the dependency of scattering amplitudes on temperature, as performed by (Rao et al., 2018).

The polarimetric radar parameters $Z_{HH}$, $Z_{DR}$ and $K_{DP}$ can be written as

$$Z_{HH} = 10\log_{10}\left[\frac{4\times10^{10}\lambda^4}{\pi^4\times|K_w|^2}\int_0^\infty \left[\left|s_{VV}^{(\pi)}\right|^2 - 2\times Re\left(s_{VV}^{*(\pi)}\left(s_{VV}^{(\pi)} - s_{HH}^{(\pi)}\right)\right)\times A_2 + \left|s_{VV}^{(\pi)} - s_{HH}^{(\pi)}\right|^2 \times A_4\right]N(D)dD\right] \quad (1)$$

$$Z_{VV} = 10\log_{10}\left[\frac{4\times10^{10}\lambda^4}{\pi^4\times|K_w|^2}\int_0^\infty \left[\left|s_{VV}^{(\pi)}\right|^2 - 2\times Re\left(s_{VV}^{*(\pi)}\left(s_{VV}^{(\pi)} - s_{HH}^{(\pi)}\right)\right)\times A_1 + \left|s_{VV}^{(\pi)} - s_{HH}^{(\pi)}\right|^2 \times A_3\right]N(D)dD\right] \quad (2)$$

$$K_{DP} = \frac{180\times\lambda\times F_{orient}}{\pi}\int_0^{D_{max}} Re\left[s_{VV}^{(0)} - s_{HH}^{(0)}\right]N(D)dD \quad (3)$$

$$Z_{DR} = \frac{Z_{HH}}{Z_{VV}} \quad (4)$$

Where $D$ (mm) is the equivalent diameter of raindrops, $\lambda$ (mm) is the radar wavelength, $s_{HH,VV}^{(*,\alpha)}$ is complex scattering amplitude at horizontal or vertical polarization for raindrops of diameter $D$, with the parameter $\boldsymbol{\alpha}$ being the angle between the incident and scattering direction (in radian, 0 for forward scattering and $\boldsymbol{\pi}$ for back scattering). Re (.) means the real part of a complex number

(Bringi and Chandrasekar, 2001; Doviak and Zrnić, 1993; Ryzhkov and Zrnic, 2019). $A_1, A_2, A_3$ and $A_4$, are angular moments for orientation of raindrop and $F_{orient}$ orientation factor which depends on the width of canting angle distribution (Ryzhkov and Zrnic, 2019). The $Z_{HH}$ and $Z_{VV}$ (dBZ) are the reflectivity factors in horizontal (both transmission and reception) and vertical (both transmission and reception) polarization, respectively.





### 3. Retrieval of DSD relations: their dependency on seasons and temperature:

### 3.1. Seasonal variation in DSD:

Earlier studies have shown large seasonal variations in DSD in southeast India and studied their impact on $Z$-$R$ relations and attenuation correction algorithms (Kozu et al., 2006; Radhakrishna et al., 2009; Rao et al., 2018, 2009, 2001; Sulochana et al., 2016). Since the present dataset is different from that of used in earlier studies (Radhakrishna et al., 2009; Rao et al., 2009, 2001), the seasonal means of $N(D)$ at different $R$ and variation of $Z$ and $D_m$ with $R$ are examined to check whether the present dataset is

able to reproduce earlier results on the seasonal behavior of DSD. Figures 1a & 1b show the variation of seasonal mean $N(D)$ with $D$ for different seasons in 2 rain rate class intervals (5-10 and 15-20 mm h$^{-1}$), respectively. The DSD exhibits clear seasonal variation at both rain rates, with smaller drops predominantly occurring during the NEM and considerable number of bigger drops during warm seasons (PRE and SWM). The observed seasonal variation corroborates earlier studies and also reaffirms that these variations are robust and characteristic features of this region. The reduction of smaller drops during the warm seasons is attributed

to the dominance of some microphysical processes, like evaporation and drop sorting, during those seasons (Radhakrishna et al., 2009).

Due to the observed large seasonal variations in DSD, the bulk rainfall parameters, like $Z$, $R$ and $D_m$ may also vary. Figures1c and 1d, respectively, show the variation of mean values of $D_m$ and $Z$ (along with standard errors) with $R$ in different seasons. The means are taken over the entire data in respective $R$ class intervals (5 mm hr$^{-1}$). As expected, clear seasonal differences are apparent in

bulk rain parameters also. Both $D_m$ and $Z$ are larger during the PRE, the hottest and convection-dominant season (Saikranthi et al., 2014), than in other seasons when $R$ is less than 60 mm hr$^{-1}$. These values are small during the NEM among all the seasons, mainly due to the presence of more (fewer) smaller (bigger) drops than in other seasons, as can be evidenced from Figure 1. The seasonal differences in bulk parameters is somewhat ambiguous at very high $R$ (>70 mmhr$^{-1}$).



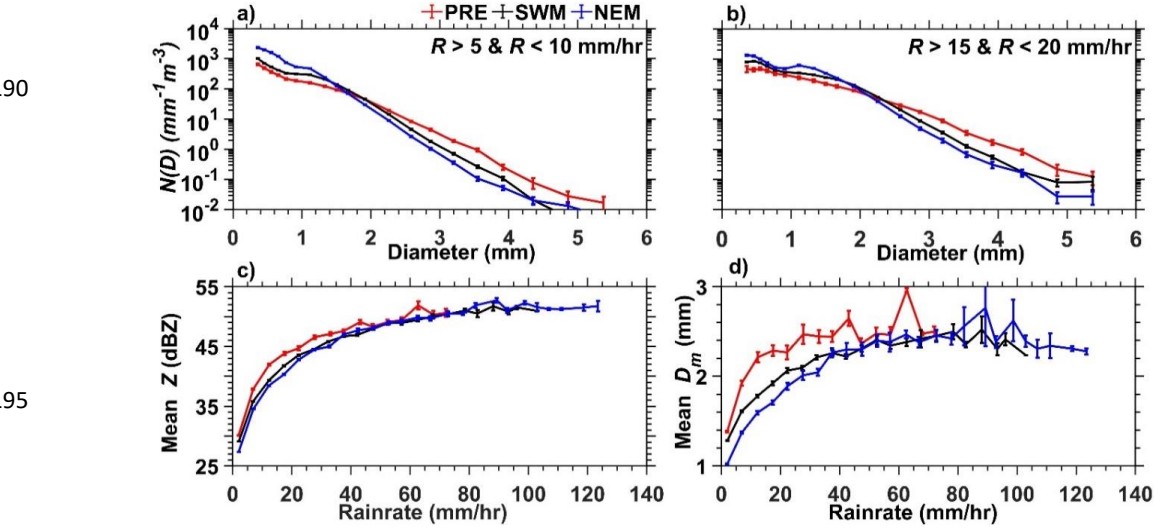

**Figure 1. seasonal mean DSD variation between the three seasons for at two rain rate intervals, i.e., (a) 5-10 and (b) 15-20 mm hr$^{-1}$. Variation of (c) mean $Z$ and (d) mean $D_m$ with $R$ during different seasons. The data within each rain rate interval**

**are averaged to obtain mean values.**





### 3.2. Retrieval of DSD relations for different seasons with various DSD models

#### 3.2.1. Exponential method

The two-parameter exponential distribution with an intercept parameter ($N_0$) and slope parameter ($\Lambda$) is the most widely used model to represent DSD in microphysical parameterization schemes and is mathematically represented as follows:

$N(D) = N_0 \exp(-\Lambda D)$ (5)

To obtain the intercept and slope parameters of the exponential distribution, first, the $D_m$ is derived from the polarimetric measurement of $Z_{DR}$ using an empirically derived relation. As $D_m$ and $\Lambda$ of the exponential distribution are related by a simple equation, $\Lambda = \frac{4}{D_m}$, the $\Lambda$ can be estimated from $D_m$. The other parameter $N_0$ is derived from $Z_H$ and the retrieved $D_m$ using another empirical relation between them (Seliga and Bringi, 1976). The most important step in this process is to derive appropriate

empirical relations between $D_m$ and $Z_{DR}$ and $Z_H/N_0$ and $D_m$, both vary with DSD and therefore are region dependent.

These empirical relations are retrieved from the scatter plots between $Z_{DR}$ and $D_m$ and log ($Z_H/N_0$) and $D_m$ (Fig. 2). Some of these parameters required for the scatter plots are computed directly from disdrometer measurements ($R$, $Z$ and $D_m$), while other polarimetric products are estimated from $T$-matrix scattering simulations (Eqs 1-4). The exponential parameters are estimated using

the method of moments, following Smith (2003). A power law fit, of the form given below, is applied on the data in Fig. 2 to obtain the coefficients in different seasons.

$D_m = a_1 Z_{DR}^{b_1}$ (6)

$D_m = a_2 \left(\frac{Z_H}{N_0}\right)^{b_2}$ (7)

where $Z_{DR}$ is represented in normal units.

Power law regression fits of the form shown in Equation 6 are fitted to the data and the coefficients (prefactor and exponent) are also shown in the figure. Good correlation is found between $Z_{DR}$ and $D_m$ in all seasons with correlation coefficients ($r^2$) of 0.9, 0.88 and 0.9 for PRE, SWM and NEM, respectively. The correlation and RMSE values during the SWM indicate that the correlation is relatively weak during that season. Although some scatter exists around the regression fits, majority of the points (as can be seen from the color bar) are close to the fit. The variance due to the scatter provides the theoretical limit on the retrieval of DSD

parameters. The coefficients of the relation change with season in accordance with the seasonal variations in DSD. From the retrieved coefficients it is clear that the $D_m$ values will be larger for the same $Z_{DR}$ during PRE and SEM than in NEM. The correlation between $Z_H/N_0$ and $D_m$ (Fig. (2d-2f)) is excellent in all seasons with an $r^2$ of 0.99. The data also closely follows the regression fits, indicating the goodness of the fit. Though the prefactor is nearly equal in all seasons, but the variation in exponent makes a difference of ~20-30% in $N_0$ value between the seasons for the same $Z_H/N_0$ and $D_m$. In other words, separate relations are

required for different seasons to reduce the uncertainty in DSD retrievals.





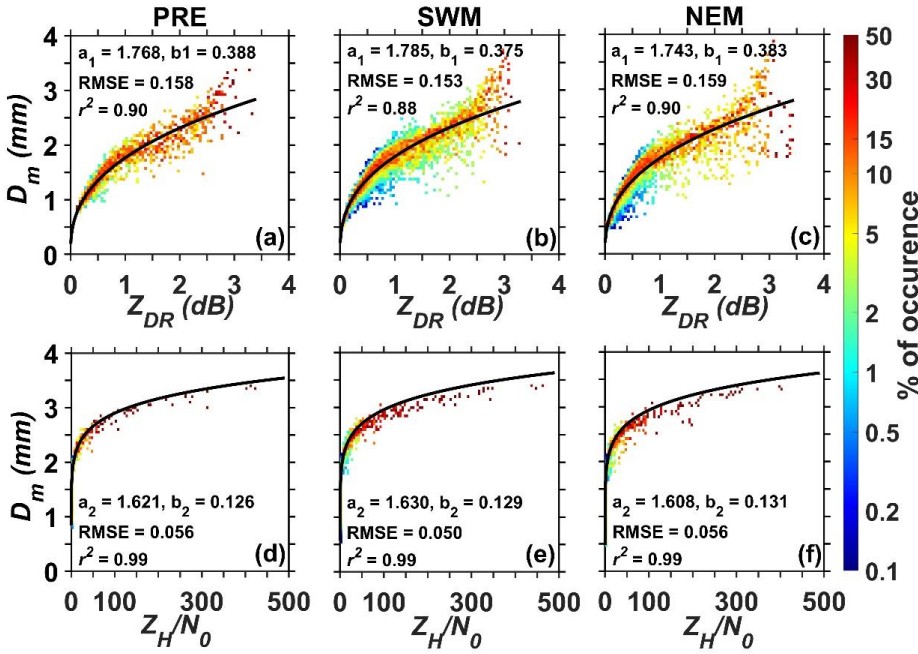

**Figure 2. Scatter plots between $Z_{DR}$ and $D_m$ for (a) PRE (b) SWM and (c) NEM seasons. (d)-(f) same as (a)-(c), but for $Z_H/N_0$ and $D_m$. The color indicates percentage occurrence of data in each cell. The power law regression fit is overlaid (solid line) on the data.**

Only a few studies exist (Gosset et al., 2010; Matrosov et al., 2005) on exponential method for the retieval of microphysical information with X-band radars. Most of the existing studies were made at longer wavelengths, at S- and C-bands. Gosset et al. (2010) obtained these power law coefficients using 11600 DSD samples collected during the AMMA field campaign in Africa. They also noted large difference in coefficients, when they retrieved with different raindrop size-shape models. The coefficients with Pruppacher and Beard (1970) model, in particular, are quite different from those obtained with other models in Africa, as seen at Gadanki. The coefficients derived at Gadanki are nearly equal to those obtained in Africa, when they are retrieved with Andsagar (1999) and Goddard (1995) models. On the other hand, Matrosov et al. (2005) noted weak dependency of coefficients on drop shape models (<6%) based on disdrometric measurements made along the west coast of United States of America, which is considered to be negligible compared to the scatter in the data used to derive the above relation.

### 3.2.2 Constrained- Gamma method

Ulbrich (1983) noted that the exponential model may not adequately represent all variations in DSD, particularly in the lower drop regime in tropical precipitation. A three-parameter gamma model is then proposed to represent all types of raindrop spectra (Ulbrich, 1983), which is expressed in the form of

$$N(D) = N_0 \, D_\mu \exp(-\Lambda D), \tag{8}$$

where, $\mu$ is the shape factor of the DSD.





To estimate three parameters of gamma distribution, three independent polarimetric variables are required. Earlier studies have shown that the three parameters of the gamma DSD model are not completely independent (Chandrasekar and Bringi, 1987; Haddad et al., 1997; Kozu and Nakamura, 1991; Ulbrich, 1983). This can be of great significance because it reduces the three parameters of gamma DSD into two parameters by constraining any two parameters, which enables us with the retrieval of DSD parameters from a pair of independent radar measurements. Zhang et al. (2001) found high correlation between $\mu$ and $\Lambda$ and proposed an empirical $\mu - \Lambda$ relation. To improve the retrieval of smaller values of $\mu$ and $\Lambda$ associated with higher rain rates, the relation was re-derived based on the truncated moment method in Brandes et al. (2003). Subsequently several $\mu - \Lambda$ relations were retrieved in different regions with varying coefficients, indicating that the $\mu - \Lambda$ relation, indeed vary with climatic regime. A new $\mu - \Lambda$ relation has been derived for monsoonal rain at Gadanki by using three years (2016-2018) of disdrometer data. The data are considered for further processing only when the drop count exceeds 1500 m$^{-3}$ and rain rate is > 5 mm hr$^{-1}$ to better retrieve values of $\mu$ and $\Lambda$ associated with higher rain rates and larger number of drops counts. The functional form of the relationship is

$$\mu = a_3 \Lambda^2 + b_3 \Lambda + c_3 \tag{9}$$

Figure 3 shows retrieved $\mu - \Lambda$ relations for PRE, SWM and NEM seasons for monsoonal rain. The $r^2$ is nearly equal among all seasons, however, the coefficients $\mu - \Lambda$ relation are found to be different for different seasons. The correlation is somewhat weaker during NEM with smaller $r^2$ and larger RMSE than in other seasons. Some scatter is also seen at higher $\mu$ and $\Lambda$ values, but their occurrence is very low. It indicates that the $\mu - \Lambda$ relation is not only region dependent, but also vary with season at the same location. The coefficients of the $\mu - \Lambda$ relation appear to be temperature dependent as we see a gradual change in coefficients from the warmest PRE to coldest NEM. Also, warmest seasons of PRE and SWM have higher slope and curvature values compared to those in NEM. It means $\mu$ will be higher during PRE and SWM than in NEM for the same $\Lambda$ for the majority of data (i.e., when $\Lambda$ and $\mu$ values are less than 8). The NEM with abundance of smaller drops with fewer bigger drops (compared to PRE and SWM) typically have smaller $\mu$ even for a larger $\Lambda$.

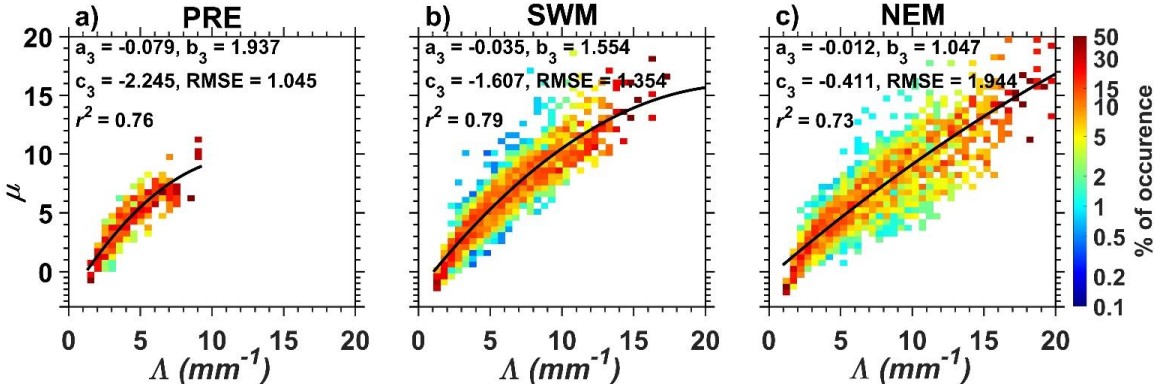

**Figure 3. Scatter plots between $\mu$ and $\Lambda$ during (a) PRE (b) SWM and (c) NEM seasons. The color indicates the percentage occurrence of data in each cell. The power law regression fit is overlaid (solid line) on the data. The statistics of regression fits are also depicted in each panel of the figure.**

As such relations are available at different locations, a comparison with them will be intuitive, which may also allow to draw some generalized conclusions. The range of curvature parameter from the published literature (Table 2) varies from 0.004 to 0.078, while the slopes and intercepts are in the range of 0.7-1.9 and 0.4-2.5, respectively. One can see that the curvature values are varying by





an order of magnitude between the regions. The differences in curvature and slope values are strikingly apparent between in warm/cold seasons/regions. The warm seasons/regions typically have larger curvature and slope values than in cold

285 seasons/regions. In fact, the smallest value of curvature (and also slope) is reported from Tibetan Plateau. Smaller values of curvature and slope are also noted during the winter monsoon season at Gadanki and in Taiwan (Seela et al., 2018). It is very clear from these comparisons that the $\mu - \Lambda$ relation is region dependent, corroborating earlier studies, but can be broadly categorized into warm and cold seasons/regions.

**Table 2: Comparison of $\mu - \Lambda$ relations obtained at Gadanki with those reported elsewhere.**

|  | Location | Seasons | $\mu - \Lambda$ relations |
|---|---|---|---|
| Present study | Gadanki, India | PRE | $\mu = -0.0788*\Lambda^2 + 1.9371*\Lambda - 2.2449$ |
|  | Gadanki, India | SWM | $\mu = -0.0383*\Lambda^2 + 1.6354*\Lambda - 1.9816$ |
|  | Gadanki, India | NEM | $\mu = -0.0117*\Lambda^2 + 1.0474*\Lambda - 0.4112$ |
| Kim et al. (2020) | Korean Peninsula | April – Oct., 2014, 2016 | $\mu = -0.01692*\Lambda^2 + 1.141*\Lambda - 2.551$ |
| Seela et al. (2018) | NCU, Taiwan | Summer | $\mu = -0.0444*\Lambda^2 + 1.549*\Lambda - 2.054$ |
|  | NCU, Taiwan | Winter | $\mu = -0.0079*\Lambda^2 + 1.019*\Lambda - 2.551$ |
| Chen et al. (2017) | Tibetan Plateau | Summer | $\mu = -0.0044*\Lambda^2 + 0.7646*\Lambda - 0.4898$ |
| Xiao et al. (2017) | Beijing | Summer (June – Sept.) | $\Lambda = 0.0194*\mu^2 + 0.7954*\mu + 2.033$ |
| Cao et al. (2008) | Oklahoma | May,2005 –May, 2007 | $\mu = -0.0201*\Lambda^2 + 0.902*\Lambda - 1.718$ |
| Brandes et al. (2003) | Florida | Summer of 1998 | $\Lambda = 0.0365*\mu^2 + 0.7354*\mu + 1.935$ |
| Zhang et al. (2001) | Florida | Summer of 1998 | $\mu = -0.016*\Lambda^2 + 1.213*\Lambda - 1.957$ |

290

Using the above retrieved $\mu - \Lambda$ relations, the gamma parameters are computed as follows. Similar to the exponential method, the $D_m$ is obtained from $Z_{DR}$ measurement. As $D_m$ is related to $\mu$ and $\Lambda$ according to the following relationship:

$$\mu = \Lambda D_m - 4 \tag{10}$$

From Eqs. 9 and 10, the following quadratic equation for $\Lambda$ is obtained

295 $$a_3\Lambda^2 + (b_3 - D_m)\Lambda + (c_3 + 4) = 0 \tag{11}$$

Solving the above quadratic equation yields two solutions for $\Lambda$ one is positive and another is negative, from which only physically possible positive $\Lambda$ value is considered. The shape parameter can be computed from the retrieved $\Lambda$ using Eq. 9. The intercept parameter $N_0$ is retrieved from radar reflectivity using the following equation (Zhang, 2017)





$$N_0 = \frac{Z_H}{\left(\frac{D_m}{4+\mu}\right)^{(7+\mu)} \times \Gamma(\mu+7)} \tag{12}$$

### 3.2.3. Normalized Gamma method:

Testud et al. (2001) proposed the normalized gamma distribution model of the form shown below to represent the DSD, which was used later in several studies (Anagnostou et al., 2008a; Tokay et al., 2020a),

$$N(D) = N_W \frac{\Gamma(4)}{3.67^4} \frac{(3.67+\mu)^{4+\mu}}{\Gamma(4+\mu)} \left(\frac{D}{D_0}\right)^{\mu} exp\left[-(3.67+\mu)\frac{D}{D_0}\right] \tag{13}$$

Where $D_0$ is the median volume diameter and $N_W$ the normalized form of intercept parameter, which is related to $D_m$ and liquid water content (LWC) as.

$$N_W = \frac{4^4\,LWC}{\pi \rho_W D_m^4} \tag{14}$$

The $D_m$ and $N_W$ can also be estimated empirically from radar parameters of $Z_H$ and $Z_{DR}$ and as follows (Tokay et al., 2020a),

$$D_m = a_4 Z_{DR}^3 + b_4 Z_{DR}^2 + c_4 Z_{DR} + d_4, \tag{15}$$

$$N_w = a_5 Z_H D_m^{b_5} \tag{16}$$

Figure 4 (a-c) shows the variation of $D_m$ with $Z_{DR}$ in PRE, SWM and NEM seasons, respectively. A third order polynomial fit of the form given in Eq. 15 has been adopted to obtain the coefficients separately for each season. Table 3 provides coefficients and fitting statistics ($r^2$ and RMSE) for each season. The variation in coefficients between the seasons is as large as 25%, indicating the strong seasonal dependency exhibited by these relations. The coefficients obtained for monsoonal rain are also different from that reported by Tokay et al. (2020) from different field campaigns (IFloodS, IPHEx and OLYMPEx). Figure 4(d-f) shows variation of log ($N_W$) with $D_m$ for PRE, SWM and NEM seasons, respectively. Coefficients for the retrieval of $N_w$ are obtained from regression fit using Eq. 16. The color in the figure represents $Z_H$ and the solid curves are obtained with retrieved coefficients for different $Z_H$ values. One can clearly see the differences in data distribution here also with considerable population at smaller $D_m$ (and larger $N_W$) during the NEM, mainly due to the preponderance of smaller drops. One can also see the near absence of smaller $D_m$ values (< 1 mm) during the premonsoon, mainly due to strong evaporation and drop sorting. These differences cause considerable seasonal variation in the retrieved coefficients (Table 3). The prefactor is found to be larger during the warmer seasons (PRE and SWM) than in colder seasons. The prefactor values are comparable to those reported by Tokay et al. (2020a) from six field campaigns.

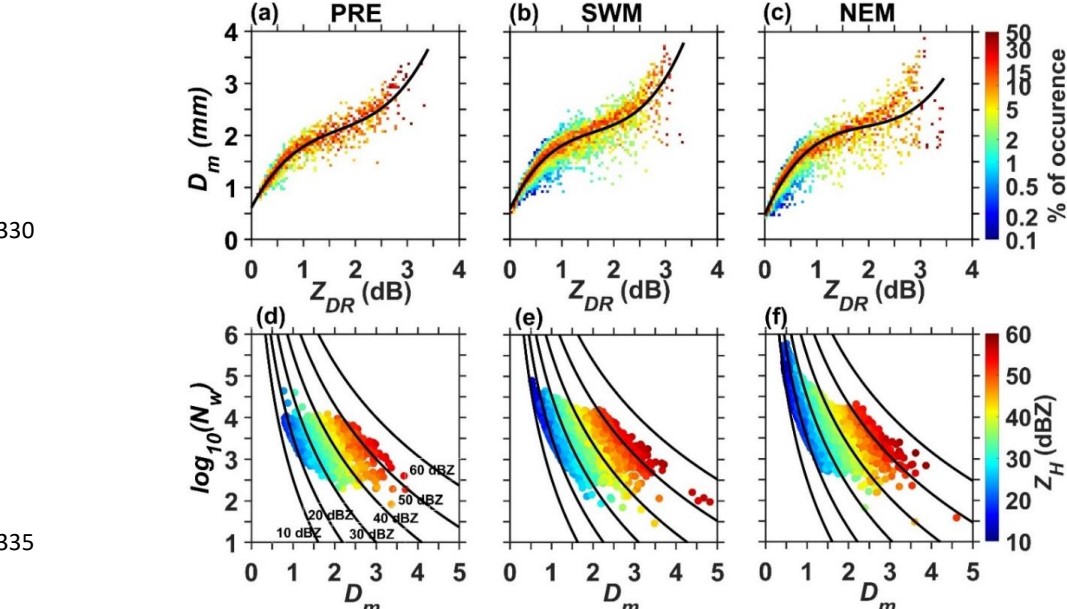

**Figure 4. Scatter plots between $Z_{DR}$ and $D_m$ for (a) PRE (b) SWM and (c) NEM seasons. Solid line is the 3rd order polynomial fit (Eq. 15). (d)-(f) Scatter plots between log $(N_W)$ and $D_m$ as a function of $Z_H$ for PRE, SWM and NEM, respectively. The solid lines indicate the variation of log $(N_W)$ with $D_m$ for different $Z_H$ values, estimated using appropriate coefficients obtained with Eq. 16.**

**Table 3: Empirically-derived coefficients of the $D_m$-$Z_{DR}$ and $N_W$-$(Z_H,D_m)$ relations for PRE, SWM and NEM seasons and statistics of curve fittings.**

|  | PRE | SWM | NEM |
|---|---|---|---|
| $D_m = a_4 Z_{DR}^3 + b_4 Z_{DR}^2 + c_4 Z_{DR} + d_4$ | | | |
| $a_4$ | 0.175 | 0.220 | 0.176 |
| $b_4$ | -0.885 | -1.068 | -1.022 |
| $c_4$ | 1.881 | 2.067 | 2.185 |
| $d_4$ | 0.614 | 0.591 | 0.497 |
| RMSE | 0.151 | 0.147 | 0.162 |
| $r^2$ | 0.91 | 0.89 | 0.90 |
| $N_w = a_5 Z_H D_m^{b_5}$ | | | |
| $a_5$ | 33.448 | 34.252 | 30.875 |
| $b_5$ | -7.380 | -7.178 | -7.185 |
| RMSE | 664 | 1094.172 | $5.36 \times 10^3$ |
| $r^2$ | 0.93 | 0.93 | 0.99 |





### 3.2.4. Beta ($\beta$) method

Most of the studies that retrieve relations between polarimetric radar products and geophysical parameters (like DSD or rain rate),
assume equilibrium drop shape model, proposed by Pruppacher and Beard (1970), which predicts an almost linear decrease of the
spheroidal raindrop aspect ratio $r$ as a function of $D$,

$r = 1.03 - 0.062 D$ (17)

where '$r$ '= $b/a$ is the axis ratio and '$b$' and '$a$' are semi minor and major axes of the rain drop, respectively (Pruppacher and Beard,

1970).The above equation gives aspect ratios close to those reported by (Pruppacher and Pitter, 1971). Drops less than about 0.5
mm were usually assumed to be spherical in shape. A number of later studies (e.g., Andsager et al., 1999; Gorgucci et al., 2001,
2000; Keenan et al. 1997) indicate that the equilibrium drop shape is not unique and the variability in drop aspect ratio–diameter
relations can be significant. The generalized form of the relation is, therefore, given as (Matrosov et al., 2002)

$r = 1.03 - \beta D$ (18)

where $\beta$ is the shape factor (mm), which is considered to be a variable rather than a fixed value by Pruppacher and Bread (1970).
It is clear that the mean shape-size relation of raindrops plays an important role in the interpretation of polarimetric radar
measurements. In order to obtain the estimator $\beta$, the $Z_H$, $Z_{DR}$, and $K_{DP}$ are used, as follows.

$\beta = a_6 \left(\frac{K_{DP}}{Z_H}\right)^{a_7} \xi_{DR}^{a_8}$ (19)

Here, the $Z_H$ is in $mm^{-6} m^{-3}$, $\xi_{DR}$ is $Z_{DR}$ in linear scale and $K_{DP}$ is in $deg\ km^{-1}$.

The $D_m$ and $N_w$ are estimated from polarimetric variables using the following equations,

$D_m = b_6 \left(\frac{\xi_{DR}-0.8}{\beta}\right)^{b_7}$ (20)

$N_w = c_6 \left(\frac{\xi_{DR}-0.8}{\beta}\right)^{c_7} Z_H^{c_8}$ (21)

The coefficients $a_{6-8}$, $b_{6,7}$, and $c_{6-8}$ of Eqs. 19-21 are derived by computing the non-linear regression analysis between each beta
and corresponding polarimetric measurements. Here, the computation has been carried out by considering the rain drops

distribution to follow a normalized gamma DSD. The intrinsic shape of the DSD is obtained by normalizing the number density
by $N_0$ (Testud et al., 2001). The retrieved coefficients in equations for $\beta$, $D_m$ and $N_w$ are given in Table 4. The mean value of $\beta$
estimated using the retrieved coefficients and Eq. 19 is in between 0.054 and 0.056 for warm seasons ~ 0.065 for NEM. The value
obtained during NEM is closer to the default value (0.062) given by Pruppacher and Beard (1970), whereas the values obtained
for PRE and SWM are much smaller, indicating that the slope of drop shape-size relation is seasonal dependent. Like other DSD

relations, the coefficients in beta method also exhibit large seasonal dependency with some of the coefficients varying by as large
as a factor of ~2.

**Table 4:  Coefficients of $D_m$ and $N_w$ retrieval equations**

|       | $a_6$ | $a_7$ | $a_8$ | $b_6$ | $b_7$ | $c_6$ | $c_7$ | $c_8$ |
|-------|-------|-------|-------|-------|-------|-------|-------|-------|
| PMON  | 1.347 | 0.385 | 1.23  | 0.338 | 0.707 | 4.628 | -0.421 | 0.072 |
| SWM   | 1.776 | 0.422 | 1.33  | 0.363 | 0.655 | 4.170 | -0.284 | 0.054 |
| NEM   | 1.902 | 0.435 | 1.43  | 0.405 | 0.580 | 4.664 | -0.283 | 0.047 |





### 3.3. Dependence of DSD relations on temperature and drop shape models

To understand the dependency of retrieved DSD relations on temperature, exponential DSD relations are considered (Eqs. 6 and 7) in this section. A temperature of 20 °C is used in the above $T$-matrix scattering simulations for computing radar parameters. To understand the dependency of retrieved coefficients on temperature, the excerise is repeated by varying temperatures from 0 °C to 30 °C in increments of 5 °C, and each time, coefficients of the above relations (Eqs. 6 and 7) are retrieved. Figure 5 shows the variation of prefactors and exponents in Eqs. 6 and 7 with temperature for different seasons. Except for $a_2$ (the prefactor in Eq.

7), all coefficients decrease monotonically with increasing temperature, albeit with different slopes. Clearly, the variation of exponent in all relations with temeprature is considerable in all seasons and is up to 6.7%, while the prefactor do not vary much with temperature and its variation is less than 2%. Among seasons, the variation in coefficients of DSD relations with temperature is larger in hot seasons than in cold season (i.e., NEM) by a factor of 2 to 6. Therefore, the variation in $D_m$ or $N_0$, for a given $Z_{DR}$ and $Z_H$, due to temperature variation is within 5% in any season, and is much less in NEM ($< 2\%$). However, the impact of seasonal

variation of coefficients on derived DSD parameters is relatively larger and is up to 20%, as discussed above.


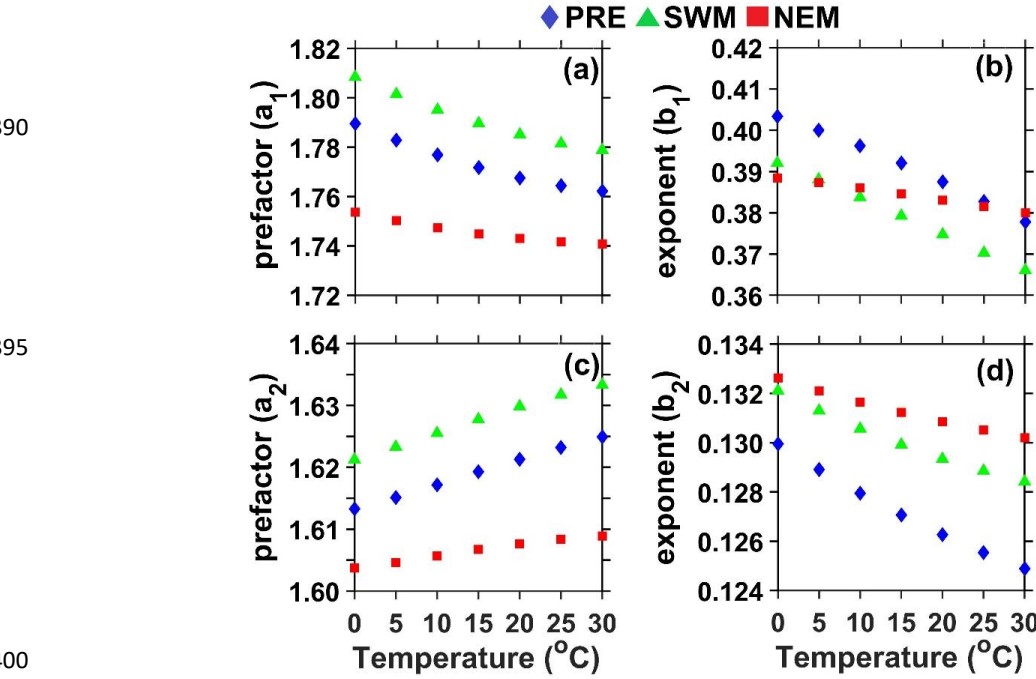


**Figure 5. Temperature dependency of coefficients of DSD parameter relations in different seasons. (a) and (b) Variation of $a_1$ and $b_1$ in Equation 6 with temperature in different seasons. (c) and (d) same as (a) and (b) but for Equation 7.**

    To examine the dependency of these coefficients on drop shape models, they are retrieved by using different drop shape models (Andsager et al., 1999; Beard and Chuang, 1987; Brandes et al., 2002; Pruppacher and Beard, 1970). The difference in coefficients

in Eq. 6 derived with different drop shape models is quite large (7-15% in prefactor and up to 28% in exponent) and in fact larger





than the seasonal difference. The prefactor (exponent) is found to be smaller (larger) with Pruppacher and Beard drop shape model than with other models. On the other hand, the dependency of coefficients in Eq. 7 on drop shape model is weak and all models yield nearly equal coefficients. The seasonal dependency of coefficients in Eq. 7 is quite high compared to their dependency on drop shape models.

**4.  Assessment of DROP-X retrieved DSD**

The degree of agreement of radar-derived DSD parameters with disdrometer-derived parameters depends on several factors: 1. The differences in sampling volumes of radar and disdrometer, 2. Vertical variability of DSD from the radar measured volume to the surface (or disdrometer measurement height) and 3. Accuracy of the empirical relations between polarimetric parameters ($Z_{DR}$, $Z_H$, $K_{DP}$) and DSD model parameters ($D_m$, $N_0$, $\mu$ and $\Lambda$). The radar sampling volume depends on the range, beam width and pulse

length. For the given radar beam width of 1°, range resolution of 150 m and a range of 450 m, the estimated sampling volume of the radar is 7264 m$^3$. To match the radar temporal resolution, the disdrometer data are averaged over 6 min (360 S).  The sampling volume of disdrometer for given surface area of 50 cm$^2$ (for JW disdrometer) and a characteristic drop size, represented by $D_m$ (or terminal velocity) of 2 mm (6.5 m s$^{-1}$) is less than 12 m$^3$. Thus, the sampling volumes differ by a factor greater than 600, which is much less than the similar comparisons made elsewhere, wherein the sampling volumes differ by a factor of 10$^5$ to 10$^7$ (Cao et al.,

2008; Tokay et al., 2020a). This is mainly due to the fact that the comparisons were made at a longer range in earlier studies. Another advantage of using shorter range for comparison studies, as is done in the present study, is the proximity of radar measuring volume to the surface. In the present study, the sampling volume is at a height of ~20 m above the disdrometer location. This reduces the bias caused by the time-height ambiguity due to the vertical variability of DSD. The retrieval accuracy also depends on empirical relations between the radar and DSD parameters, as these relations vary with season (as shown in Section 3). However,

appropriate relations have been used for comparison in the present study to reduce such ambiguity.

Evaluation of DROP-X derived DSD parameters, using retrieved coefficients in different DSD formulations discussed above, is carried out by comparing them with those derived with disdrometer observations.  For this purpose, disdrometric dataset during 2019-20, which has not been used for the retrieval of coefficients, is used for comparison. Long duration events (longer than 2 hours) are selected for the evaluation of DSD retrieval techniques. A total of 6 events each from SWM and NEM are selected for

this purpose (Table 5).  These events include a variety of precipitating systems, including thunderstorms and mesoscale convective systems.



**Table 5: Details of rain events (date, duration, number of radar samples within the event and type of event) used for assessment of four DSD retrievals.**

| Season | Date | Duration (HH:MM) | Number of radar Samples | Type Of Rain |
|---|---|---|---|---|
| SWM | 17/08/2019 | 08:01 | 74 | MCS |
| | 20/08/2019 | 06:00 | 58 | MCS/ISLT |
| | 11-12/09/2019 | 03:23 | 33 | MCS |
| | 12-13/09/2019 | 03:05 | 30 | MCS |
| | 15/09/2019 | 02:57 | 27 | ISLT |
| | 16/09/2019 | 03:08 | 30 | ISLT |
| NEM | 04/10/2019 | 03:35 | 33 | ISLT |
| | 30-01/11-12/2019 | 04:01 | 34 | MCS |
| | 11/10/2020 | 04:06 | 35 | MCS |
| | 22-23/10/2020 | 04:33 | 41 | ISLT |
| | 15/11/2020 | 02:04 | 20 | MCS |
| | 15/11/2020 | 01:52 | 19 | MCS |

### 4.1 Case studies

Figure 6 shows variation of rainfall bulk parameters during two precipitation events, one each from SWM (on 12 September 2019) and NEM (on 15 November 2020), chosen as case studies. It also shows typical spatial maps of $Z_H$ observed with DROP-X during the passage of precipitating system on the above days.  On 12 September 2019, a convective cell originated southwest of the study region at 16:00 IST and has grown quickly into a mesoscale storm with leading convective and trailing stratiform region. It propagated eastward and passed the radar location around 22:00 IST as an intense storm stretched in north-south direction. The DROP-X has tracked this storm, when it passed over the radar site. The DROP-X measured $Z_H$is in the range of 50-52 dBZ during the storm's passage across the radar site at 22:00 IST. The collocated disdrometer also shows $Z$ as large as 52 dBZ and a rain rate of 38 mm hr$^{-1}$ at the time of passage of the core of the storm. The disdrometer-estimated $D_m$ is also found to be large (2.7 mm) at that time (Figure 6). Light-moderate rain with $Z$, $R$ and $D_m$ in the range of 23-38 dBZ, 0.5-5 mm hr$^{-1}$ and 1-2 mm, respectively, continued for about 3 hours after the passage of this intense convective cell over the radar site.

The second case study is from the NEM occurred on 15 November 2020. The NEM was active on the day with wide spread clouds over the southeast peninsular India. A rain band of width ~40 km stretched in southwest-northeast direction moved northwestward and produced widespread rainfall over the study region for about 2 ½ hours.  Rain intensity is light to moderate during the above period with $R$ always less than 5 mm hr$^{-1}$ and $Z_H$ varying in the range of 10 – 40 dBZ. The disdrometer-derived $D_m$ is also found to be small (1 - 2 mm) during the above period.





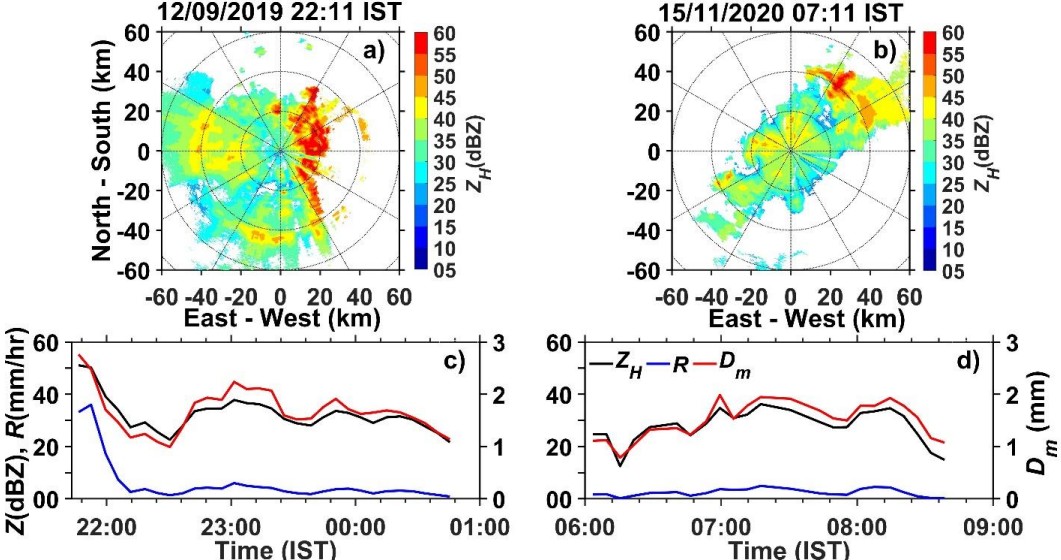

**Figure 6. Spatial variation of $Z_H$ measured by DROP-X on (a) 12 September 2019 and (b) 15 November 2020. (c) and (d)**

**Temporal variation of rainfall bulk parameters ($Z_H$, $R$ and $D_m$) measured by disdrometer on the above dates, respectively.**

The $D_m$, shape and slope parameters of different DSD models estimated from DROP-X measurements using retrieved coefficients (Section 3) are compared with those obtained with disdrometer in Figure 7. The $D_m$ values obtained by Exponential, CG and N-gamma are equal and are superposed on each other, while those derived with β method differ from the above methods. In general, $D_m$ values obtained by all methods show good correspondence with that derived by the disdrometer. In particular, all methods well

capture the peaks in the $D_m$ variation. However, the temporal variation of $D_m$ by β method shows more and larger spikes relative to the reference, in particular on 12 September 2019 (Figure 7a). It is expected that the noisy $K_{DP}$ and $Z_{DR}$ at lower rain rates leads to large error in the estimation of β (Gorgucci et al., 2002). However, Figures 6 and 7 show that the disagreement between the β method and disdrometer- and other-radar derived $D_m$ is significant even at moderate to high rain rate ($R > 5$ mm hr$^{-1}$). Anagnostou et al. (2008a) also noted such large differences by β method during convective regimes in their DSD retrieval assessment study. In

addition, Anagnostou et al. (2008b) noted a gradual increase in uncertainty in retrieved DSD parameters and is attributed to inadequate attenuation correction. The disdrometer location in the present study is very near to the radar (~200 m) and, therefore, attenuation (and correction) is negligible. On the other hand, the observed differential phase, supposed to represent the differential propagation phase, is contaminated with differential backscattered phase in the presence of strong convection (Trömel et al., 2013). Adaptive Kalman filtering is used in the present study to smooth out the fluctuations and differential backscattered phase, which

is found to be very effective in removing the above affects. However, some uncertainty remained in the removal of differential backscattered phase when strong convection occurs close to the radar location. It could be the reason for the small bias in $D_m$ by techniques based on $K_{DP}$.

As expected (given that there is a good agreement in $D_m$ by radar and disdrometer and the relation $\Lambda = \frac{4}{D_m}$), the temporal variation of radar-derived $\Lambda$ by exponential method matches well with that of disdrometer in both cases (Figures 7c and 7d). Though the

temporal variation of $\Lambda$ and $\mu$ by CG method matches reasonably well with those obtained with disdrometer, their magnitudes differ from the reference data, particularly overestimation of both parameters is noted on 12 September 2019 case.

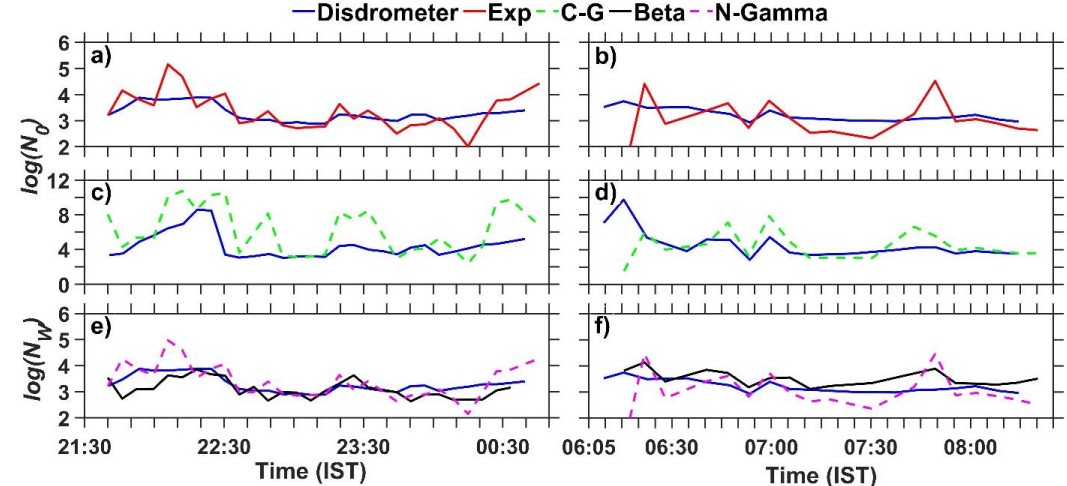

**Figure 7: Comparison of (a) and (b) $D_m$, (c) and (d) $\Lambda$ by assuming exponential distribution, (e) and (f) $\Lambda$ by assuming gamma distribution, and (h) $\mu$ by assuming gamma distribution on 12 September 2019 and 15 November 2020, respectively, with disdrometer derived values.**

The temporal variations of log $N_0$ with Exp and CG methods and log $N_W$ with N-Gamma and $\beta$ methods along with that of disdrometer are shown in Fig. 8. The agreement with reference is generally good for log $N_W$ by $\beta$ and N-Gamma methods. The $N_0$ values obtained with Exp method also agrees reasonably well with those obtained by disdrometer. However, the agreement is poor with CG method and it generally overestimates log $N_0$ values relative to disdrometer values, mainly due to the overestimation of $\mu$. Except for CG method, all root mean square error (RMSE) between the retrieved and reference $N_0/N_W$ is $\leq 1$.

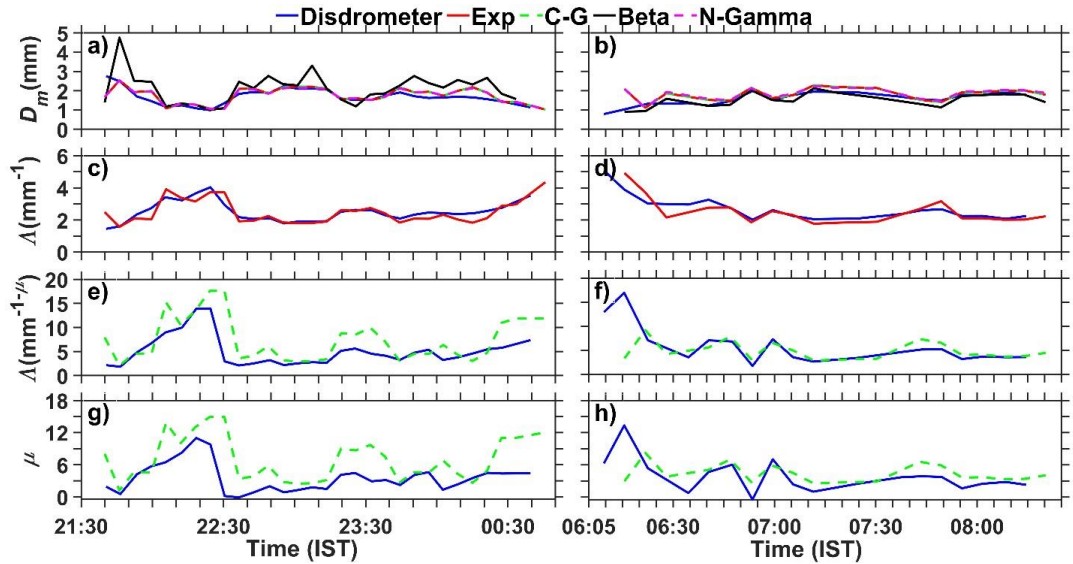

**Figure 8. Comparison of log $N_0$ (a) and (b) by assuming exponential distribution and (c) and (d) by assuming gamma distribution on 12 September 2019 and 15 November 2020, respectively, with disdrometer-derived log $N_0$. (e) and (f) Comparison of log $N_W$ by N-Gamma and $\beta$ methods with disdrometer-derived log $N_W$ on the above days.**



**4.2. Statistical assessment**

As shown in Table 4, data from six long events each from SWM and NEM are used to assess the radar-derived $D_m$ and $N_0/N_W$ against those obtained with disdrometer. These events include a variety of precipitation systems, from isolated thunderstorms to mesoscale scale convective systems. Fig. 9 shows the statistical comparison of $D_m$ and $N_0/N_W$ derived by radar (4 methods) and disdrometer for all the events given in Table 6. The colored symbols in each scatter diagram represent the data from different seasons (green solid triangle -SWM and red open square - NEM). Table 2 summarizes different comparison statistics of four retrieval methods under testing for SWM and NEM seasons. Clearly, the statistical comparison also shows that the comparison is better for the retrieval of $D_m$ than $N_0/N_W$ by all methods. All methods show correlation better than 0.65 ($r^2$) and RMSE less than 0.55. Among $D_m$ retrieval by different methods, $\beta$ method shows better correlation than others in both seasons, but suffers with large RMSE values. The distribution of data is also wider in case of $\beta$ method. The agreement between radar retrievals and disdrometer-derived $D_m$ is relatively better during the NEM than in SWM. On the other hand, the retrieval of $N_W$ by N-Gamma method is much better in both seasons compared to other methods. The CG method shows weaker correlations and larges RMSE values than other methods, mainly because of the problems related to $K_{DP}$ and $\mu$.

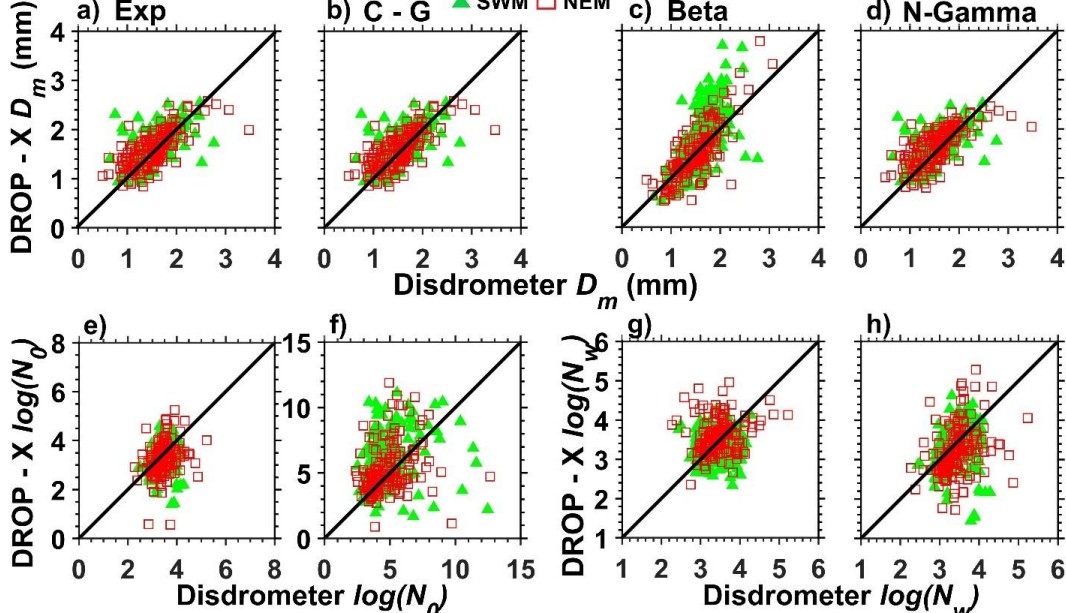

**Figure 9. Scatter plots of $D_m$ obtained by disdrometer and DROP-X with (a) exponential (b) constrained gamma, (c) $\beta$ and (d) normalized gamma methods for SWM (solid green triangle) and NEM (open red square) seasons. (e)-(h) same as (a)-(d) but for log $N_0/N_w$**

... 


**Table 6: Evaluation statistics of $D_m$ and log $N_0/N_w$ by Exp, CG, and N-Gamma $\beta$ methods for SWM and NEM.**

| Parameters | Statistics | SWM | | | | NEM | | | |
|---|---|---|---|---|---|---|---|---|---|
| | | Exp. | C-G | Beta | N-Gamma | Exp. | C-G | Beta | N-Gamma |
| $D_m$ | $r^2$ | 0.65 | 0.65 | 0.68 | 0.65 | 0.71 | 0.71 | 0.81 | 0.69 |
| | Bias | -0.06 | -0.06 | -0.17 | -0.12 | -0.06 | -0.06 | -0.02 | -0.09 |
| | RMSE | 0.29 | 0.29 | 0.55 | 0.29 | 0.32 | 0.32 | 0.38 | 0.34 |
| $log (N_0)$ or $log (N_w)$ | $r^2$ | 0.37 | 0.20 | 0.32 | 0.46 | 0.23 | 0.21 | 0.46 | 0.63 |
| | Bias | 0.16 | -0.78 | 0.12 | 0.20 | -0.31 | -1.24 | 0.20 | 0.18 |
| | RMSE | 0.55 | 2.15 | 0.49 | 0.50 | 1.08 | 2.73 | 0.50 | 0.70 |

## 5. Summary and conclusion

Five years of disdrometric measurements and 2 years of DROP-X measurements have been used, for the first time, to i) obtain relations for the retrieval of DSD parameters appropriate for monsoonal rain and study their dependency on temperature and drop
size – shape relations, ii) understand the seasonal variation of coefficients and iii) assess the DROP-X-derived DSD by various DSD retrieval methods. Using 3 years of disdrometer-measured DSD, various polarimetric parameters have been computed using $T$-matrix simulations. Coefficients of four commonly used DSD relations are retrieved empirically from simulated data. Important results emanated from the study are summarized as follows.

1. The coefficients for obtaining DSD parameters by exponential, CG, N-Gamma and $\beta$ methods for monsoonal rain are found to
be different from other regions, indicating that they are region dependent. The mean value of $\beta$ estimated at Gadanki is closer to the default value (0.062) given by Pruppacher and Beard (1970) during the NEM, whereas the values obtained for PRE and SWM are much smaller, indicating that the slope of drop shape-size relation is season dependent and 0.062 is more applicable for colder season. To understand the dependency of the coefficients of these relations on temperature and drop shape models, the coefficients of Exp method are retrieved for different temperatures and drop shape models. It is found that the variation in
$D_m$ or $N_0$, for a given $Z_{DR}$ and $Z_H$, due to temperature variation is within 5% in any season, and is much less in NEM ($< 2\%$). However, the dependency of coefficients in $D_m - Z_{DR}$ equation on drop shape model is high (7-15% in prefactor and 28-28% in exponent) and in fact is higher than on seasons. The dependency of coefficients on drop shape models is found to be different in different geographical regions. While the dependency is found to be high at Gadanki and in Africa, it is found to be weak along the west coast of United States of America.

2. The present study corroborates some of the earlier studies that showed the $\mu - \Lambda$ relation is region dependent. It clearly shows that this relation is also season and temperature dependent, as we see a gradual change in coefficients from the warmest PRE to coldest NEM. Also, warmest seasons of PRE and SWM have higher slope and curvature values compared to those in NEM. It means $\mu$ will be higher during PRE and SWM than in NEM for the same $\Lambda$ for the majority of data (i.e., when $\Lambda$ and $\mu$ values are less than 8). A comparison of $\mu - \Lambda$ relations obtained in different seasons at Gadanki with those available in the literature
elsewhere clearly reveals that warm seasons/regions typically have larger curvature and slope values than in cold seasons/regions.



3. The disdrometer data clearly shows large seasonal variation with preponderance of smaller drops during NEM compared to warm seasons, corroborating earlier findings (Rao et al. 2001; 2009; Radhakrishna et al. 2009). As a result, the obtained coefficients also show large seasonal variation. From the retrieved coefficients it is clear that the $D_m$ values will be larger for the same $Z_{DR}$ during PRE and SEM than in NEM. Though the prefactor is nearly equal in all seasons, but the variation in exponent makes a difference of ~20-30% in $N_0$ value between the seasons for the same $Z_H/N_0$ *and $D_m$*. Among seasons, the variation in coefficients of DSD relations with temperature is larger in hot seasons than in cold season (i.e., NEM) by a factor of 2 to 6. However, the impact of seasonal variation of coefficients on derived DSD parameters is relatively larger and is up to 20%. Therefore, appropriate coefficients need to be used while retrieving DSD from polarimetric measurements.

4. The four commonly used radar retrieval methods of DSD are evaluated with the help of two case studies (one each from SWM and NEM) and data from 12 events. All methods retrieve $D_m$ reasonably well and produce high correlation and small RMSE against the reference. The $\beta$ method alone produced wide range of $D_m$ values similar to that of disdrometer. However, the scatter is large, particularly in convection mainly due to the fact the comparison is made close to the radar site, where the differential phase is often contaminated by differential backscattering phase. As a result, the RMSE exhibited by $\beta$ method is also found to be large. Comparison of retrievals of $N_0/N_W$ with those of disdrometer shows the superiority of N-Gamma method over other methods. All other methods compare poorly with disdrometer-derived $N_0/N_W$ with small $r^2$ and large RMSE values. Considering all the factors (Table 4), N-Gamma method is found to be better in retrieving the DSD parameters. However, such assessment studies are also planned at longer ranges (10 km and 35 km) with DROP-X to understand the strengths and limitations of the above methods in retrieving DSD accurately.

**Data Availability**

The data used in the present study belongs to National Atmospheric Research Laboratory and can be obtained on request.

**Author contribution**

**KA:** Data curation, Writing- Original draft preparation, Data analysis, Software; **TNR**: Conceptualization, Supervision, Manuscript Editing. **NRR**: Supervision and Editing, **KAJ**: Software and Editing,

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
