# Peer review of "Retrieval of Microphysical Parameters of Monsoonal rain Using X-band Dual-polarization Radar: Their Seasonal Dependence and Evaluation"

_Atmospheric Measurement Techniques, 2022_

## Referee Comment (RC1)

**REVIEW REPORT**

Review of amt-2022-291

By Kumar Abhijeet, T. Narayana Rao, N. Rama Rao, and K. Amar Jyothi

Manuscript Title – Retrieval of Microphysical Parameters of Monsoonal rain Using X-band Dual-polarization Radar: Their Seasonal Dependence and Evaluation

**GENERAL COMMENTS**

The manuscript analyzed F5 years of disdrometer data and 2 years of radar data in India to obtain relation for the DSD retrievals from radar data. The influences of different factors on these relations have been analyzed and the comparison between disdrometer DSD parameters and radar based DSD parameters has been performed. The paper is well written and well organized. In my opinion in a bit log and some sections can be very shortened or eliminated (such as section 4.1 and 3.3).

I suggest the publication after addressing my comments:

- 1. Line 58: I think that also the generalized gamma by Thurai et al. (2018) needs to be added in this list.
- 2. Line 84: Probably also Italy needs to be added in this list.
- 3. Line 140: can the authors justifies the choice of 6 minutes of integration?
- 4. Line 150: "...simulation with other models..." which other models? Please clarify this sentence
- 5. Lines 184-185: this conclusion is true for Dm but much less marked for Z in particular at high rain rates.
- 6. Lines 229-230: It should be highlighted that in some cases the differences among the coefficients are very limited. For example, between a1 of Dm-Zdr relation for PRE e NEM. It should be interesting to define the error in terms of Dm in using only one relation for all the seasons.
- 7. Table 4: Probably this is PRE not PMON
- 8. Please note that the Authors need to change Figure 7 with Figure 8 and vice versa.

**REFERENCE**

Thurai, M., & Bringi, V. N. (2018). Application of the generalized gamma model to represent the full rain drop size distribution spectra. Journal of Applied Meteorology and Climatology, 57(5), 1197-1210.

---

## Referee Comment (RC2)

**Review of amt-2022-291**

This is a review of " **Retrieval of Microphysical Parameters of Monsoonal rain Using X-band Dual-polarization Radar: Their Seasonal Dependence and Evaluation**" by Kumar Abhijeet, T. Narayana Rao, N. Rama Rao, and K. Amar Jyothi, submitted to Atmospheric Measurement Technique

This paper deals with an interesting topic which is relevant to the community using Doppler radars for the estimation of rainfall and retrieving rain microphysical parameters thereby unravelling the structure of mesoscale convective systems.

The manuscript discusses different approaches for retrieving microphysical parameters from DROP-X radar and compared the results with a surface-based disdrometer. The manuscript is written well, and easy to follow. I strongly feel that the results are worth publishing in Atmospheric Measurement Technique (AMT). This topic is important to a variety of disciplines and scientists, I strongly recommend the paper for publication in AMT after addressing the following minor comments/suggestions.

Minor comments and Suggestions:

Page 2: line 47: remove; after Ryzhkov and Zrnic, 2019;

Page 3: Line 92: Include Lavanya et al. 2019

Page 6: line 188: 70 mmhr-1 (space between mm hr-1)

Page 6: Include details of standard error bars in the Figure1 caption.
Page 6: Figure 1 caption: Correct "seasonal …." with "Seasonal …."

Page 13: Table 4: Replace PMON with PRE

Page 18: Captions of Figure 7 and Figure 8 are interchanged.
Accordingly please correct them in the text.

Page 17: line 467: I think the authors are referring to Figure 8. (Dm, shape and slope parameters are shown) (Please see my above comment)

---

## Author Comment (AC1)

**Response to reviewer 1 comments**

The manuscript analyzed 5 years of disdrometer data and 2 years of radar data in India to obtain relation for the DSD retrievals from radar data. The influences of different factors on these relations have been analyzed and the comparison between disdrometer DSD parameters and radar based DSD parameters has been performed. The paper is well-written and well organized. In my opinion in a bit log and some sections can be very shortened or eliminated (such as section 4.1 and 3.3).

*We would like to thank the reviewer for appreciating our work and providing suggestions to improve the readability and quality of manuscript. We have implemented these minor suggestions and also shortened Section 4.1 in the revised manuscript.*

Comment: 1. Line 58: I think that also the generalized gamma by Thurai et al. (2018) needs to be added in this list.
*Reply: The suggested reference (Thurai et.al. 2018) is added in the revised manuscript.*

Comment: 2. Line 84: Probably also Italy needs to be added in this list.
*Reply: Sorry for not adding Italy. It is now added.*

Comment: 3. Line 140: can the authors justifies the choice of 6 minutes of integration?
*Reply: The JW disdrometer provides the DSD measurements at 1-minute temporal resolution. However, one volume scan of radar takes ~6 minutes. Therefore, disdrometer data are averaged over 6 minutes to match to the temporal resolution of the radar for faithful comparison.*

Comment: 4. Line 150: "…simulation with other models…" which other models? Please clarify this sentence
*Reply: What we mean by other models are, Pruppacher and Beard, 1970; Beard and Chuang, 1987; and Brandes et al., 2002. As these model names are listed in the preceding sentence, the sentence is modified in the revised manuscript as "Though simulations with Andsager et al. (1999) model are finally used in our analysis, simulations with other raindrop size-shape models mentioned above are also performed to check the dependency of scattering amplitudes and retrieved polarimetric radar parameters on drop shape model."*

Comment: 5. Lines 184-185: this conclusion is true for Dm but much less marked for Z in particular at high rain rates.
*Reply: Yes. It is true. That is why, it is clearly mentioned in the manuscript that the seasonal differences are clear and prominent at rain rates less than 60 mm $hr^{-1}$.*

6. Lines 229-230: It should be highlighted that in some cases the differences among the coefficients are very limited. For example, between a1 of Dm-Zdr relation for PRE e NEM. It should be interesting to define the error in terms of Dm in using only one relation for all the seasons.
*Reply: It is indirectly mentioned in the text. However, as per reviewers' suggestion, it is explicitly mentioned in the revised manuscript. Also, as per reviewers' suggestion, the errors due to the*

*usage of one relation/existing relations elsewhere are estimated (Figure A). It is found that the error in $D_m$ due to the usage of single relation is high in SWM (among three seasons). The mean error is around 6%, however, on occasions, the error is as large 30%. However, the error in $D_m$ due to utilization of existing relations in the literature (reported at other locations) is very large with the mean error as large as 15%. Most importantly, the variability is quite large and at times the error is >100%.*

[Figure]

*Figure A: Percentage error in $D_m$ due to the usage of single $Z_{DR}$-$D_m$ relation (derived using total data) compared to seasonal relations (SWM – Southwest monsoon; NEM – Northeast monsoon and PRE – Premonsoon) and the existing relations elsewhere (BR-Brandes et al., 2004; CAO- Cao et al., 2008; MAT – Matrosov et al. 2005). For this analysis, 4 years of 1 min. JWD-measured DSDs have been used.*

7. Table 4: Probably this is PRE not PMON
*Reply: It is changed now as PRE*

8. Please note that the Authors need to change Figure 7 with Figure 8 and vice versa.
Figure 7 is changed to Figure 8 and Figure 8 is changed to Figure 7.
*Reply: Sorry for the mistake. It is now corrected in the revised manuscript.*

---

## Author Comment (AC2)

**Response to reviewer 2 comments**

This paper deals with an interesting topic which is relevant to the community using Doppler radars for the estimation of rainfall and retrieving rain microphysical parameters thereby unravelling the structure of mesoscale convective systems.

The manuscript discusses different approaches for retrieving microphysical parameters from DROP-X radar and compared the results with a surface-based disdrometer. The manuscript is written well, and easy to follow. I strongly feel that the results are worth publishing in Atmospheric Measurement Technique (AMT). This topic is important to a variety of disciplines and scientists.

I strongly recommend the paper for publication in AMT after addressing the following minor comments/suggestions.

*We would like to thank the reviewer for strongly recommending the paper for publication with minor revision and also providing suggestions to improve the readability and quality of manuscript. We have implemented these minor suggestions in the revised manuscript.*

**Minor comments and Suggestions:**

Comment: **Page 2: line 47: remove; after Ryzhkov and Zrnic, 2019;**
*Reply: corrected*

Comment: **Page 3: Line 92: Include Lavanya et al. 2019**
*Reply: The suggested reference (*Lavanya et al. 2019*) is added in the revised manuscript.*

Comment:**Page 6: line 188: 70 mmhr-1 (space between mm hr-1)**
*Reply: Corrected*

Comment:**Page 6: Include details of standard error bars in the Figure1 caption.**
*Reply: Details of standard error bars are included in Figure 1 caption.*

Comment:**Page 6: Figure 1 caption: Correct "seasonal …." with "Seasonal …."**
*Reply: Corrected.*

Comment:**Page 13: Table 4: Replace PMON with PRE**
*Reply: It is changed now as PRE*

Comment:**Page 18: Captions of Figure 7 and Figure 8 are interchanged.
Accordingly please correct them in the text.
Page 17: line 467: I think the authors are referring to Figure 8. (Dm, shape and slope parameters are shown) (Please see my above comment)**

*Reply: Sorry for the mistake. It is now corrected in the revised manuscript.*